# A Taylor's power law in the Wenchuan earthquake sequence with
# fluctuation scaling
**Peijian Shi[1], Mei Li[2*], Yang Li[3*], Jie Liu[2], Haixia Shi[2], Tao Xie[2], Chong Yue[2]**
[1]Co-Innovation Center for Sustainable Forestry in Southern China, College of Biology and the
Environment, Bamboo Research Institute, Nanjing Forestry University, Nanjing 210037, China
[2]China Earthquake Networks Center, China Earthquake Administration, Beijing 100045, China
[3]Department of Mathematics and Statistics, University of Minnesota Duluth, Duluth, MN 55812, USA
*Correspondence to*: mei_seis@163.com (M. Li); yangli@d.umn.edu (Y. Li)
**Abstract**   Taylor's power law (TPL) describes the scaling relationship between the
temporal or spatial variance and mean of population densities by a simple power law.
TPL has been widely testified across space and time in biomedical sciences, botany,
ecology, economics, epidemiology, and other fields. In this paper, TPL is analytically
reconfirmed by testifying the variance as a function of the mean of the released
energy of earthquakes with different magnitudes on varying timescales during the
Wenchuan earthquake sequence. Estimates of the exponent of TPL are approximately
2, showing that there is mutual attraction among the events in the sequence. On the
other hand, the spatial–temporal distribution of the Wenchuan aftershocks tends to be
nonrandom but approximately definite and deterministic, which highly indicates a
stable spatial–temporal dependent energy release caused by regional stress adjustment
and redistribution during the fault revolution after the main shock. Effect of different
divisions on estimation of the intercept of TPL straight line has been checked while
the exponent is kept to be 2. The result shows that the intercept acts as a logarithm
function of the time division. It implies that the mean–variance relationship of the
energy release from the earthquakes can be predicted although we cannot accurately
know the occurrence time and locations of imminent events.

**1  Introduction**
The Wenchuan $M_S$ 8.0 earthquake on May 12, 2008 was the result of the
intensively compressive movement between the Qinghai–Tibet Plateau and the
Sichuan basin. It ruptured the middle segment of the Longmenshan (LMS) thrust belt
(Burchfiel et al., 2008), with a total length of fault trace of approximately 400 km
along the edge of the Sichuan basin and the eastern margin of the Tibetan plateau, in
the middle of the north–south seismic belt of China. Millions of aftershocks have
occurred after the main event. Up to now, the focus zone tends to be quiet with only
small ones occurring occasionally. A complete Wenchuan earthquake sequence has
been attained.
Statistical seismology applies statistical methods to the investigation of seismic
activities, and stochastic point process theory promotes the development of statistical
seismology (Vere–Jones et al., 2005). After some improvement, most of the
point process theories and methods can be used to analyze spatio–temporal data of
earthquake occurrence and to describe active laws of aftershocks. The term
"aftershock" is widely used to refer to those earthquakes which follow the occurrence
of a large earthquake and aggregately take place in abundance within a limited
interval of space and time. This population of earthquakes is usually called an
earthquake sequence. In seismological investigations, one important subject has long
been the statistical properties of the aftershocks. Spatial and temporal distribution of
aftershocks after a destructive earthquake is usually performed in a general survey
(Utsu, 1969). In seismology, one of the most famous theories describing the activities
of aftershocks is the Gutenberg–Richter law (Gutenberg and Richter, 1956), which
expresses the relationship between the magnitude and the total number of earthquakes
with at least that magnitude in any given region and time interval. Another one is the
Omori's law, which was first depicted by Fusakichi Omori in 1894 (Omori, 1894) and
shows that the frequency of aftershocks decreases roughly with the reciprocal of time
after the main shock. Utsu (1969) and Utsu et al. (1995) developed this law and
proposed the modified Omori formula afterwards. Since the 1980s, with the
development of nonlinear theory, an epidemic–type aftershock sequence (ETAS)
model has been proposed by Ogata (1988, 1989, 1999), which is based on the
empirical laws of aftershocks and quantifies the dynamic forecasting of the induced
effects. This model has been used broadly in earthquake sequence study (Kumazawa
and Ogata, 2013; Console, 2010).

An increasing number of investigations show that there is an interaction effect for

the occurrence of aftershocks in a given area. Stress triggering model is usually used
to depict interaction between larger earthquakes by the view of physics (Haris, 1998;
Stein, 1999). More and more results show that obvious enhancement in Coulomb
stress not only promotes the occurrence of upcoming mid or strong events of an
earthquake sequence but also affects their spatial distribution to some degree
(Robinson and Zhou, 2005).
The goal of this paper is to introduce a different statistical method called Taylor's
power law into the statistical seismology field by analyzing the Wenchuan earthquake
sequence from the point of view of energy distribution or energy release. We aim to
find out whether or not the energy distribution or energy release of the Wenchuan
earthquake sequence complies with a specific power–law function of TPL for
different scaled samples, and what the spatial and temporal properties are.

In statistics, there are two important moments in a distribution, the mean ($\mu$)

and the variance ($\sigma^2$). It is common to describe the types of the distributions using the
relationship between these two parameters. For instance, we have $\sigma^2 = \mu$ for a Poisson
distribution. In nature, however, the variance is not always equal to or proportional to
the mean. Mutual attraction or mutual repulsion for individuals in natural populations,
e.g., the intra–specific completion of plants, makes variance different from the mean.
After examining many sets of samples of animal and plant population spatial densities,
Taylor (1961) found that the variance appears to be related to the mean by a power–
law function: the variance is proportional to the mean raised to a certain power

$$\sigma^2 = a\mu^b \tag{1}$$

or equivalently as a linear function when the mean and variance are both
logarithmically transformed

$$\log_{10}(\sigma^2) = \log_{10}(a) + b \times \log_{10}(\mu) = c + b \times \log_{10}(\mu) \tag{2}$$

where $a$ and $b$ are constants and $c = \log_{10}(a)$. Eqs. 1 or 2 is called Taylor's power law
(henceforth TPL) or Taylor's power law of fluctuation scaling (Eisler et al., 2008).

Eqs. 1 and 2 may be exact if the mean and variance are population moments

calculated from certain parametric families of skewed probability distributions
(Cohen and Xu, 2015). TPL describes the species–specific relationship between the
spatial or temporal variance of populations and their mean abundances (Kilpatrick and
Ives, 2003). It has been verified for hundreds of biological species and abiotic
quantities in biomedical sciences, botany, ecology, epidemiology, biomedical sciences,
botany, and other fields (Taylor, 1961, 1984; Kendal, 2002; Eisler et al., 2008; Cohen
and Xu, 2015; Shi et al., 2016, 2017; Lin et al., 2018). Most of the scientific
investigations of TPL mainly focus on the power–law exponent $b$ (or slope $b$ in the
linear form), which has been believed to contain information on aggregation in space
or time of populations for a certain species (Horne and Schneider, 1995).
In this study, we also concentrate on the parameter $b$ of TPL. We expect that $b$ is
independent of the temporal block size $A$ which is used to divide the Wenchuan
sequence into different temporal blocks because the aftershock area is invariable
during this period.
**2   Wenchuan earthquake sequence**

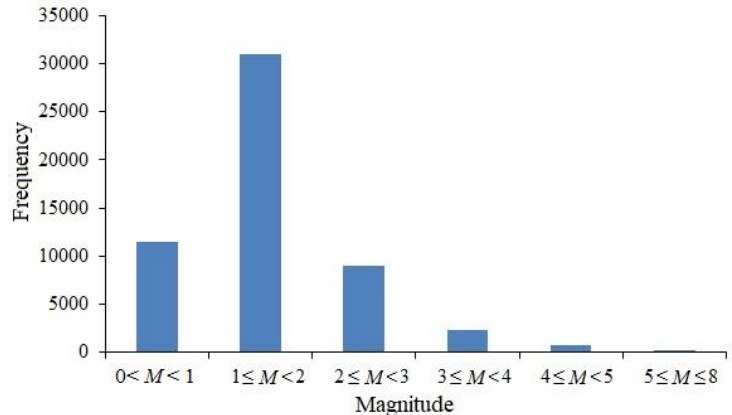


**Figure 1.** Histogram of earthquakes with different magnitudes of the Wenchuan sequence.

A large earthquake of magnitude $M_S$ 8.0 hit Wenchuan, Sichuan province of

China at 14:28:01 CST (China Standard Time) on May 12, 2008 with an epicenter

located at 103.4 °N and 31.0 °E and a depth of 19 km.

According to the earthquake catalogue of the China Earthquake Networks Center

(CENC) (http://www.csi.ac.cn/), there have been 54,554 earthquakes of magnitudes

$M > 0$ recorded for the Wenchuan sequence by December 31, 2016. Figure 1 shows

the frequency of aftershocks with different magnitudes. Here, aftershocks with $M <$

2.0 account for 77.9% of the total sequence due to the fact that only weak ones occur

after a long period of time after the main shock. In addition, except for the main shock,

the number of aftershocks is 733 for magnitudes $4.0 \leqslant M < 5.0$, and 86 for $5.0 \leqslant$

$M < 8.0$, respectively. They account for a very small percentage of the total.




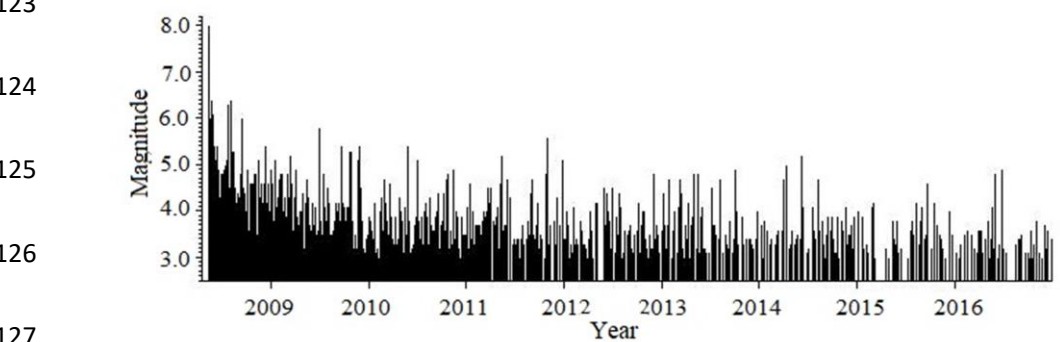


**Figure 2.** Series plot of the Wenchuan earthquake sequence with $M \geqslant 3.0$ from May 12, 2008 to
December 31, 2016.

Figure 2 displays the fluctuation variability of the Wenchuan earthquake

sequence with $M \geqslant 3.0$ from May 12, 2008 to December 31, 2016. The temporal

distribution of the magnitudes of aftershocks attenuates quickly after the main shock.

The three larger aftershocks all occurred in 2008 with *M* 6.4 on May 25, *M* 6.1 on
August 1, and *M* 6.1 on August 5, respectively. Eighty–five percent of aftershocks
with $M \geqslant 3.0$ occurred by the end of 2011, about 2.5 years after the main shock.

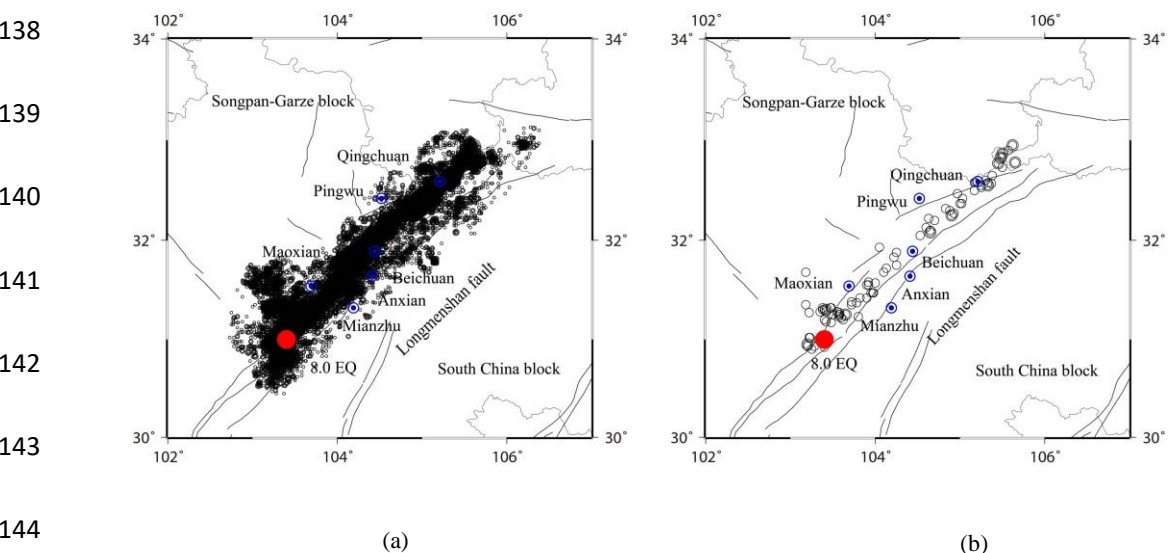

(a)                        (b)

**Figure 3.** Spatial distribution of epicenters of the Wenchuan earthquake sequences with (a) *M* > 0 and (b) $M \geqslant 5.0$ from May 12, 2008 to December 31, 2016.The main shock on May 12, 2008 is labeled by a red solid circle.

Figure 3a shows the spatial distribution of epicenters of the Wenchuan
earthquake sequence with *M* > 0 from May 12, 2008 to December 31, 2016. The
aftershocks are distributed in the region with latitude 102°E–107°E and longitude
30°N–34°N, mainly along the Longmenshan thrust fault, which is a junction region of
Songpan–Garze block and South China block and extends along north–east–east
(NEE) direction for more than 400 km. The size of the aftershocks on different scales
is characterized by a population density of the events distributed in space and time
after the Wenchuan $M_S$ 8.0 earthquake but we neglect the variations of the aftershock
area in the next step. The distribution of strong aftershocks is of different segment
characteristics. Earthquakes with magnitude $M \geqslant 5.0$ mainly spread in south
Miaoxian and Mianzhu area and north Pingwu area. There are no strong aftershocks
occurring in the middle areas such as Beichuan and Anxian (see Figure 3b).
According to the primary investigation results of the Wenchuan rupture process
conducted by Chen et al. (2008), the rupture of the Wenchuan 8.0 earthquake
originated from Wenchuan thrust fault with a little right lateral slip component and
extended mainly in north–east (NE) orientation. The whole process formed two areas
with larger dislocations. One is the south area of Miaoxian located in the bottom
section in Figure 3b. The other one lies near Beichuan area (the middle segment in
Figure 3b) but no strong shocks happened there.

**3   Data processing method**
For the complete Wenchuan earthquake sequence, we denote the number of all
earthquakes by $N$, i.e., $N = 54,554$, and use $q = 1, \ldots, N$ to index each earthquake. For
each earthquake with magnitude $M_q$, its corresponding energy release is labeled by $E_q$
and it can be attained in the light of the following relationship (Xu and Zhou, 1982)
$$\log_{10}\left(E_q\right) = 11.8 + 1.5M_q \tag{3}$$
We use $t_q$ to index the time lag of the $q$–th aftershock from the main shock (in
days), i.e., $t_1 = 0$ for the main event. The last aftershock occurred at 18:05:57 CST
(China Standard Time) on December 31, 2016, and its $t_q$ value is 3155.
In order to study the relationship between the variance and mean of the energy
sequence $E_q$, we first divide it into equally–spaced short temporal blocks with size $A$
(in days). For example, if $A = 10$, then the number of blocks is $N/A = 3155/10 = 315.5$
which is rounded to the nearest integer. Now the complete energy sequence $E_q$ is
partitioned into $n = 316$ blocks of short energy subsequences. We use $i$ to index each
block, i.e., $i = 1, \ldots, n$ and $h_i$ to denote the number of data points in each block which
is variable because earthquakes occurred stochastically in the sequence. Now we can
calculate the mean $(\mu)$ and variance $(\sigma^2)$ for each block using
$$\mu_i = \frac{\sum_{j=1}^{h_i} E_{i,j}}{h_i} \tag{4}$$
$$\sigma_i^2 = \frac{\sum_{j=1}^{h_i} (E_{i,j} - \mu_i)^2}{h_i - 1} \tag{5}$$
where $E_{i,j}$ denotes the energy of the $j$–th earthquake in the $i$–th block.

**4   Results**

The data processing procedure has been performed with different block size $A =$

4, 5, 6, …, 100. The number of sample points in each block decreases as the block
size increases. The relationships between the mean and variance of the released
energies from earthquakes in 6 representative temporal blocks are shown in Figure 4
on a log–log scale. The red line stands for the fitted linear function of TPL's power
law $\log_{10}(\sigma^2) = c + b \log_{10}(\mu)$ using least squares. The 95% confidence intervals (CI)
of the slope and the coefficients of determination $R^2$ are shown in Table S1. For
instance, Figure 4a shows the variance as a function of the mean for 316 time
intervals when $A = 10$. The estimated intercept is 0.702 and the estimated slope is
2.060 with 95% CI (1.989, 2.076), and $R^2 = 0.963$. The root-mean-square error
(RMSE) was also calculated to exhibit the feasibility of using a TPL with the
exponent 2 to approximate that with the exponent to be estimated (unknown).

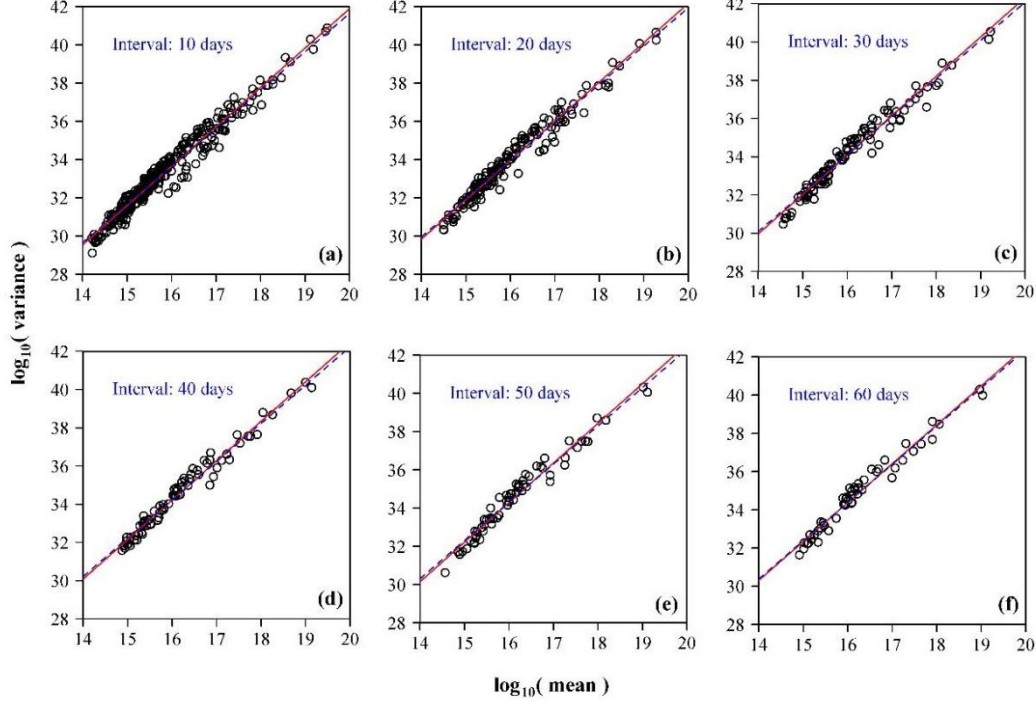


**Figure 4.** Calculated variance as a function of the observed mean of the energies from
earthquakes in each time interval on a log–log coordinate (open circles), for different values of $A$. The
red straight line corresponds to the fitted Taylor's power law with an unknown exponent, i.e. $\log_{10}(\sigma^2) =$
$c + b \log_{10}(\mu)$, using least squares. The blue dashed line corresponds to the fitted Taylor's power law
with the exponent 2, i.e. $\log_{10}(\sigma^2) = d + 2 \log_{10}(\mu)$. There are 97 different values of $A$ in total, and only
6 are shown here. (a) $A = 10$; (b) $A = 20$; (c) $A = 30$; (d) $A = 40$; (e) $A = 50$; and (f) $A = 60$.

Figure 4 and Table S1 show that there is an apparent linear relationship between
the common logarithm of the variance and that of the mean for all earthquakes
occurring within different temporal blocks, characterized by a property of aggregation
on different timescales. The estimated value of the intercept, $c$ (or $\log_{10}(a)$), which is
mainly influenced by the number of samples, overall increases with $A$ from 0.016 to
3.249 (Table S1). The estimates of slope $b$, on the other hand, are roughly 2 for all
block sizes used in the study. All $R^2$ values are greater than 0.96, showing a very
strong linear relationship. These results indicate that the energy release of aftershocks
of the Wenchuan sequence complies well with a temporal TPL.

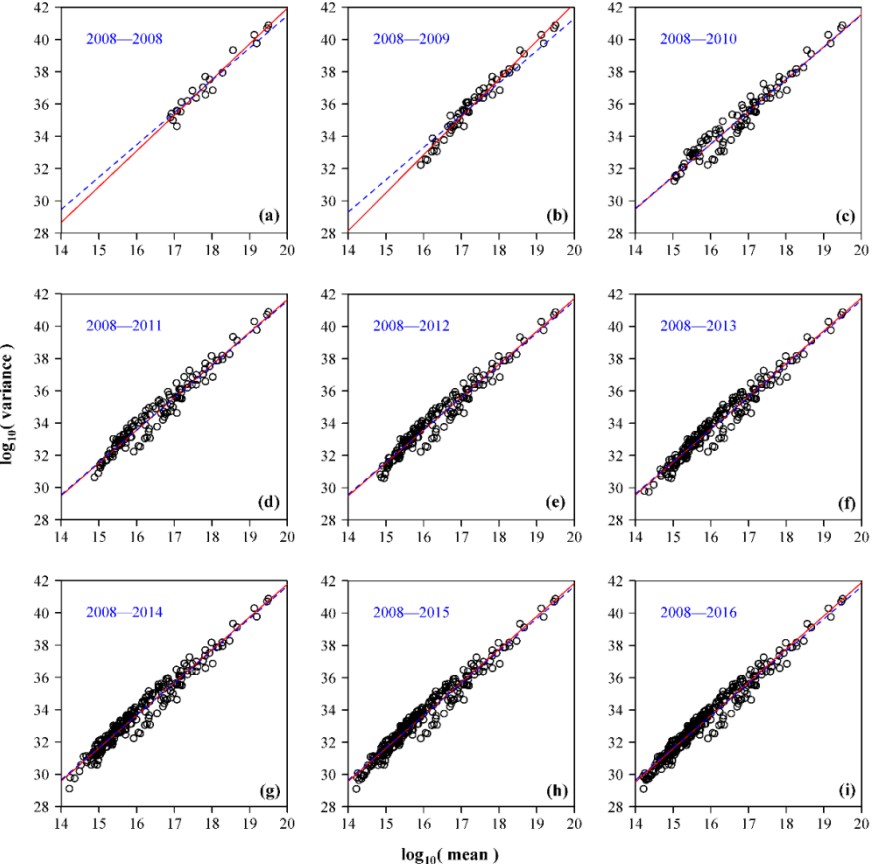


**Figure 5.** The calculated variance as a function of the observed mean of the energies from earthquakes
in each block on a log–log scale (open circles) when $A$ is fixed to be 10. The red straight line
corresponds to the fitted Taylor's power law with an unknown exponent, i.e. $\log_{10}(\sigma^2) = c + b \log_{10}(\mu)$,
using least squares. The blue dashed line corresponds to the fitted Taylor's power law with the
exponent 2, i.e. $\log_{10}(\sigma^2) = d + 2 \log_{10}(\mu)$. (a) 2008–2008; (b) 2008–2009; (c) 2008–2010; (d) 2008–
2011; (e) 2008–2012; (f) 2008–2013; (g) 2008–2014; (h) 2008–2015; and (i) 2008–2016.


Next, we divide the Wenchuan earthquake sequence into 9 time stages in years:
2008–2008, 2008–2009, 2008–2010, 2008–2011, 2008–2012, 2008–2013, 2008–2014,
2008–2015, and 2008–2016. For each stage, we follow a similar procedure leading to
Figure 4. That is, we first transform all earthquakes into their energy forms using the
relationship between earthquake magnitude $M$ and energy $E$. Then the energy
sequence are partitioned into temporal blocks with a fixed block size $A = 10$ days. The
calculated variances and means are plotted on a log-log scale as shown in Figure 5.
Again, TPL comes into play for all time stages. The estimates of the parameters in Eq.
(2) for the data in different stages were listed in Table S2.

Figure 5 shows a strong linear relationship between the variance and mean of

the earthquake energy populations on a log-log scale, especially for those large
samples. The estimates summarized in Table S2 (red fitted lines in Figure 5) show
similar results as in Table S1. The intercept gradually increases as the total number of
samples increases but with a little more fluctuation. Meanwhile, the estimate of slope
$b$ is still roughly a constant around 2.

Here with the exponent $b = 2$ considered, the possible relationship between the

estimate of the intercept (namely $d$) in equation $\log_{10}(\sigma^2) = d + 2 \log_{10}(\mu)$ and the
temporal block size $A$ is also examined. The estimated intercepts of the Wenchuan
sequence as $A$ increases from 4 days to 100 days in 1−day increments are shown in
Figure 6. At the same time, a logarithm function and an exponential function are
employed respectively to fit the data (i.e., $d = \alpha + \beta \times \log_{10}(A)$ and $d = m \times A^n$, where
$\alpha$, $\beta$, $m$ and $n$ are constants), and the results show that the logarithm function has a
higher goodness of fit (namely a lower residual sum of squares). The estimate of
parameter $\alpha$ is equal to 0.7398 with 95% CI (0.7246, 0.7581), and the estimate of
parameter $\beta$ is equal to 0.9121 with 95% CI (0.9004, 0.9229). Because $\log_{10}(a) = d =$
$\alpha + \beta \times \log_{10}(A)$, we will have:
$$\sigma^2 = a\mu^2 = 10^{\alpha} A^{\beta} \mu^2 \tag{6}$$
It illustrates that the variance of energy releases from aftershocks depends on two
factors: (i) the mean squared, and (ii) the size of temporal block defined. Up to now,
we confirm that the mean−variance relationship of energy releases from an
earthquake sequence can be quantified although the accurate prediction of the time
and location of an imminent event is still not attainable.

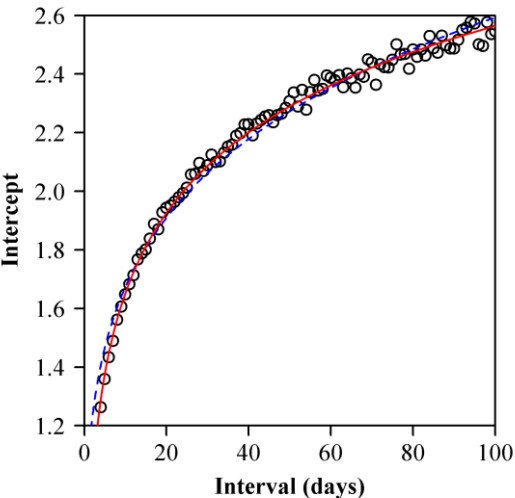


Figure 6 The effect of time division (time span) on the estimate of the intercept in the TPL with a fixed
exponent of 2, i.e. $\log_{10}(\sigma^2) = d + 2 \log_{10}(\mu)$, where $d$ denotes the intercept. Two equations were used to
fit the data ($d = \alpha + \beta \times \log_{10}(A)$ and $d = m \times A^n$, where $\alpha$, $\beta$, $m$ and $n$ are constants). The residual sum
of squares (= 0.0535) using the logarithm function (represented by the red curve) is lower than that (=
0.1460) using the exponential function (represented by the blue curve).

**5 Discussion**
The evolutionary process of a large earthquake is characterized by some complex
features from stochastic to chaotic or pseudo–periodic dynamics (McCaffrey, 2011).
On the one hand, there is a long–term slow strain of accumulation and culminating of
rocks in the rigid lithosphere prior to the event with a sudden rupture and
displacement of blocks. On the other hand, there is another long–term slow strain of
redistribution and energy release with a large number of aftershock occurrences in an
extensive area, which generally lasts for several months, sometimes even years, after
the main shock.
It has been statistically established that in populations, if individuals distribute
randomly and are independent of each other, then the variance is equal to the mean,
i.e., $\sigma^2 = \mu$; individuals show mutual attraction if the variance is proportional to the
mean to a power $> 1$, i.e., $\sigma^2 > \mu$; individuals mutually repel each other if the
variance is proportional to the mean to a power $< 1$, i.e., $\sigma^2 < \mu$ (Taylor, 1961; Horne
and Schneider, 1995). The results obtained in this study show that the exponent of the
TPL is around 2 in the Wenchuan energy sequence either with different time span $A =$
4, 5, 6, …, 100 days or with a fixed time span $A = 10$ days but for 9 time stages
between 2008 and 2016. This means earthquakes in the Wenchuan sequence are not
distributed at random and independent of each other but with a mutual attraction. It
also indicates that there are possible interactions among different magnitudes in the
earthquake sequence. Cohen and Xu (2015) proposed analytically that observations
randomly sampled in blocks from any skewed frequency distribution with four finite
moments give rise to TPL because the variation in the sample mean and sample
variance between blocks are theoretically small if every block is randomly sampled
from the same distribution.
There are various types of interpretations for the value of parameter $b$. Ford and
Andrew (2007) suggested that individuals' reproductive correlation determines the
size of $b$. While Kilpatrick and Ives (2003) proposed that interspecific competition
could reduce the value of $b$. Above all, empirically, $b$ usually lies between 1 and 2
(Maurer and Taper, 2002). However, it is expected that TPL holds with $b = 2$ exactly
in a population with a constant coefficient of variation (CV) of population density.
This expectation derives from the well–known relationship: SD (standard deviation)
equals to square root of variance ($\sigma^2$), i.e., $SD = \sigma$ and the coefficient of variation
$CV = SD/\mu = k$, here $k$ is a constant. Then we can obtain $\sigma^2 = (k\mu)^2$. The relationship
between $\log_{10}(\sigma^2)$ and $\log_{10}(k\mu)$ is a straight line with slope 2 on a log–log scale.
It is well established that there is a specific property on the population either in
space or in time when $b$ equals 2. Ballantyne (2005) proposed that $b = 2$ is a
consequence of deterministic population growth. While Cohen (2013) showed that $b =$
2 arose from exponentially growing, noninteracting clones. Furthermore, using the
Lewontin–Cohen (LC) model of stochastic population dynamics, Cohen et al. (2013)
provided an explicit, exact interpretation of its parameters of TPL. They proposed that
the exponent of TPL will be equal to 2 if and only if the LC model is deterministic; it
will be greater than 2 if the model is supercritical (growing on average) and be less
than 2 if the model is subcritical (declining on average). This property indicates that
parameter $b = 2$ in our investigation on the Wenchuan earthquake sequence depends
exactly on its specific distribution of aftershocks. In other words, the law of
occurrence of all events or energy release in space and time is deterministic following
the main shock on May 12, 2008.
Although various empirical confirmations suggest that no specific biological,
physical, technological, or behavioral mechanism can explain all instances of TPL,
there has been some improvement in understanding the distribution and duration time
of aftershocks after the main event. Jiang et al. (2008) studied the Wenchuan
earthquake sequence using Gutenberg–Richter law (Gutenberg and Richter, 1956) and
the Omori's law (Omori, 1894). Their investigation attained a specific relationship
between the magnitude and the total number of earthquakes for a stable $b$ value,
which indicates that the frequency of aftershocks decreases roughly with the
reciprocal of time after the main shock. One of the models with physical parameters is
the stress triggering mechanism put forward by Dieterich (1994) with Dieterich and
Kilgore (1996). Shen et al. (2013) achieved a good fit between the observed
Wenchuan aftershocks and the analytic solution of the modified Dieterich model.
Their results suggested that the generation of earthquakes is actually related to the
state of fault and can quantitatively describe the temporal evolution of the aftershock
decay. In this sense, the Wenchuan energy sequence satisfies TPL with slope $b = 2$,
indicating a stable spatial–temporal dependent energy release caused by regional
stress adjustment and redistribution during the fault revolution after the main shock.
These results are of high coherence with what has been attained by Christensen et al.
(2002), who proposed a unified scaling law linking together the Gutenberg–Richter
Law, the Omori Law of aftershocks, and the fractal dimensions of the faults. Their
results show that nonzero driving force in the crust of the Earth leads to an earthquake
as a sequence of hierarchical correlated processes and this mechanism responsible
for small events also is responsible for large events. In other words, a main shock and
an aftershock are consequences of the same process.

It is possible that there are some interactions among earthquakes with different

magnitudes in an earthquake sequence. This kind of interaction probably derives from
medium stress state of the focus zone where earthquakes happen. The stress field in
the aftershock area is in a rapidly adjusting state when a lager earthquake occurred. It
is probable that a light stress adjustment caused by a small earthquake most likely
induces an obvious event in its surroundings in the near future. This process can lead
to aggregation of aftershocks in space and time in extensive areas, causing TPL to
hold for the Wenchuan earthquake energy sequence. However, whether TPL accords
with all earthquake sequences and complies with specific parameters, e.g., $b = 2$,
needs further investigation. Up to now, one thing we can confirm is that the missing
events can lead to the exponent in TPL increase. For example, the estimated $b$ is
approximately 2.1 to 2.2 if the events with magnitude $M > 1.0$ are used. It indicates
missing events can change the state of energy release from a stable (deterministic)
state to an unstable (supercritical) state as Cohen et al. (2013) have proposed.

The current study shows that the exponents of TPL for different temporal blocks

for the Wenchuan earthquake sequence are approximately equal to 2 universally. The
estimated intercept could be expressed as a linear equation of the log-transformation
of temporal block $A$ (Figure 6). The goodness of fit of the nonlinear regression is
fairly high ($R^2$ = 0.9940 in Figure 6), indicating some interesting underlying

mechanism leading to the occurrence of the aftershocks. The distribution of the energy releases from aftershocks should be a right-skewed unimodal curve that can be reflected by magnitude frequency distribution as shown in Figure 1. In fact, Cohen and Xu (2015) have demonstrated that the correlated sampling variation of the mean and variance of skewed distributions could account for TPL under random sampling and the estimated exponent of TPL was proportional to the skewness of the distribution curve. For an exponential distribution, the variance equals its mean squared. However, in our study, although the variance of energy releases from aftershocks is similarly proportional to its mean squared, the coefficient of proportionality (i.e., $a$ in Eq. [1]) does rely on the size of the temporal block. This means that the energy releases from aftershocks might follow a temporal block-dependent generalized exponential distribution, which should be more complex than the generalized exponential distribution (Gupta and Kundu, 2007). However, the distribution function for the energy releases from aftershocks has not been well defined so far. The existing functions for describing a skewed distribution of energy releases or magnitudes usually belong to pure statistical models that lack clear physical dynamic mechanism. Our study suggests that further studies should focus on a temporal block-dependent or a sub-region-dependent distribution. However, to provide a clear mathematical expression for this distribution function is beyond the topic of this paper. It deserves further investigation.

## 6 Conclusions

In summary, we attempt to use a new way to investigate a spatio–temporal

distribution property of aftershocks of the Wenchuan earthquake sequence during
2008–2016. In terms of the energy release, the variance of samples in the earthquake
population is shown to have a simple power law relationship as a function of the mean
on different timescales, which gives rise to a TPL, i.e., $\sigma^2 = a\mu^b$, with $b = 2$. On the
one hand, the results show that the intercept of the fitted line in linear form $\log_{10}(\sigma^2)$
$= c + b \times \log_{10}(\mu)$ on a log–log scale, increases as the number of samples and it is
reconfirmed that parameter $c$ (namely $\log_{10}(a)$) predominantly depends upon the size
of the sampling units (Taylor, 1961). On the other hand, if TPL holds, the estimated
values of parameters $a$ and $b$ support the conclusion that the Wenchuan aftershocks
mutually trigger each other and distribute in space and time not randomly but
determinantly and definitely. We fix the exponent of TPL to be 2, and check the
effects of different time divisions on the estimate of the intercept. The result shows
that the intercept acts as a logarithm function of the timescale. It implies that the
mean–variance relationship of energy releases from the earthquakes can be predicted
even though we cannot accurately predict the time and location of imminent events.

**Acknowledgments**   The work has been funded from NSFC (National Natural
Science Foundation of China) under grant NO. 41774084 and National Key R&D
Program of China under grant NO. 2018YFC1503506. P.S. was supported by the
Priority Academic Program Development of Jiangsu Higher Education Institutions.

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
