# Peer review of "A Taylor's power law in the Wenchuan earthquake sequence with fluctuation scaling"

_Natural Hazards and Earth System Sciences, 2018_

## Referee Comment (RC1) · Anonymous Referee #1 · 26 Dec 2018

Ms. No.: nhess-2018-315 Title: A Taylor's power law in the Wenchuan earthquake sequence with fluctuation scaling

General: This manuscript applied Taylor's power law (TPL) into the released energy of earthquakes in the Wenchuan aftershocks. It confirms the existence of TPL in earthquake sequence, i.e. the variance is shown as a power law function of the mean. TPL holds for different time spans, although the intercept values of the linear regressions increase with the increase of time spans. I think this is a good paper that is well written and scientifically interesting. I applaud the general approach and central research question of the paper. I did not find any major problem. Here I listed my suggestions which may help in improving the manuscript.

1. I suggested incorporating section '2 Taylor's power law' into 'Introduction', and

adding a new section 'Data source and processing method' (or similar subhead) which includes section '1 Wenchuan earthquake sequence' and part of section '3 Data processing method and results'. Finally I would expect four sections in the main text: 'Introduction', 'Data source and processing method', 'Results', and 'Discussion and conclusions'.

2. I think lines 181-182 can be deleted as they are duplicated with lines 178-179.

3. It is not clear for me why the authors placed Figures 5 and 6 into Discussion but not Results. Would it be better to move them into Results?

4. It is interesting that the authors showed the positive relationship between the intercept and the interval (Figure 6). Inspired by them, I drew the relationship between the slope and the interval (see attached figure, panel a). Furthermore, the interval shows strong negative relationship with the slope (panel b). This is interesting and I have no idea to explain them. I understand this may be out of the scope of the current manuscript, so the authors do not need to discuss them here. This may be helpful in their future work.
* * *

[Figure]

[Figure]

**Fig. 1.**

---

## Referee Comment (RC2) · Anonymous Referee #2 · 18 Jan 2019

**Review of the paper**

**A Taylor's power law in the Wenchuan earthquake sequence with fluctuation scaling**

by **Peijian Shi et al.**

The authors explore Taylor's power law (TPL) of the released energy of Wenchuan earthquakes sequence on varying timescales. In my opinion the paper is interesting and well written. Here there are a couple of major comments:

- The authors show that the exponent of the TPL is a value independent of timescale equal or almost equal to 2. Why does it happen? The authors mention some general explanations, some of them related to Ecology, but none of them is focused on the released energy of earthquakes along time. In other works, my question is, for what physical reason would the exponent of the TPL be equal to 2?
- Note that using the logarithm to model the intercept as a function of the temporal block size A, the overall formula of the TPL remains,
$$V = kA^{\beta}\mu^{b}$$
This is an interesting expression that would deserve more attention in the paper. In particular, the exponent of A (it is around 0.2) is an interesting parameter that should be highlighted and discussed in the paper.

**Minor comments**

- Lines 20-22: "On the other hand…approximately definite and deterministic". This paragraph is not clear, what does it mean?
- I realize that the authors are using along the paper logarithms with base 10, it should be mentioned. Moreover, notation "lg" for logarithms is not standard and it would be better changed to "log" along the paper.
- Usually, when the mean is denoted with the Greek letter $\mu$ the variance is denoted as $\sigma^2$.
- Lines 296-298: "The variations of the estimated exponent b… are shown in Figure 6". Something is wrong here because Figure 6 shows the variation of the estimated intercept.

---

## Author Comment (AC2) · 13 Feb 2019

mei\_seis@163.com

Received and published: 13 February 2019

We thank the reviewers for their comments. Answers are given below in red. Changes in the revised version of the paper are also in red.

**Anonymous Referee #2**

A Taylor's power law in the Wenchuan earthquake sequence with fluctuation scaling by Peijian Shi et al. The authors explore Taylor's power law (TPL) of the released energy of Wenchuan earthquakes sequence on varying timescales. In my opinion the paper is interesting and well written. Here there are a couple of major comments: - The authors show that the exponent of the TPL is a value independent of timescale equal or almost equal to 2. Why does it happen? The authors mention some general explanations,

some of them related to Ecology, but none of them is focused on the released energy of earthquakes along time. In other works, my question is, for what physical reason would the exponent of the TPL be equal to 2? Considering the previous investigations, we think the physical reason of the slope b=2 in TPL during our work is that, after the main shock, there is a stable spatial-temporal dependent energy release caused by regional stress adjustment and redistribution during the fault revolution after the main shock. The related words are also added to the revised paper in red.

- Note that using the logarithm to model the intercept as a function of the temporal block size A, the overall formula of the TPL remains, V =  $kA\beta\mu b$  This is an interesting expression that would deserve more attention in the paper. In particular, the exponent of A (it is around 0.2) is an interesting parameter that should be highlighted and discussed in the paper. We have added something respectively to Section results and discussion in red.

Minor comments - Lines 20-22: "On the other hand...approximately definite and deterministic". This paragraph is not clear, what does it mean? - I realize that the authors are using along the paper logarithms with base 10, it should be mentioned. Moreover, notation "Ig" for logarithms is not standard and it would be better changed to "log" along the paper. We have changed "Ig" into "log10" in revised paper.

- Usually, when the mean is denoted with the Greek letter  $\mu$  the variance is denoted as  $\sigma$ 2. We have changed "V" into " $\sigma$ 2 " in revised paper.

- Lines 296-298: "The variations of the estimated exponent b... are shown in Figure 6". Something is wrong here because Figure 6 shows the variation of the estimated intercept. Yes and the reviewer is right. We have changed the sentence "The variations of the estimated exponent b of the Wenchuan sequence... are shown in Figure 6" into "The variations of the estimated intercept of the Wenchuan sequence... are shown in Figure 6".

Please also note the supplement to this comment: https://www.nat-hazards-earth-syst-sci-discuss.net/nhess-2018-315/nhess-2018-315-AC2-supplement.pdf

СЗ

---

## Referee Comment (RC3) · Anonymous Referee #3 · 14 Feb 2019

Earthquake forecasting is very important but challenging. Due to the inherent randomness and complexity of rupture process, the forecast can only be made in probabilistic manners rather than in the form of deterministic predictions. Therefore, statistical methods play an important role in earthquake forecasting and hazard assessment. Taylor's power law (TPL) has been widely testified across space and time in biomedical sciences, botany, ecology and many other fields. The current manuscript shows its application in the study of Wenchuan earthquake sequence. The results suggest that the mean–variance relationship of the energy release from the earthquakes could be predicted and the exponent of TPL are approximately 2.

The manuscript is well organized. The discussion is thorough and the methods are solid. I recommend that the manuscript could be considered for publication after reply-
ing the following questions.

1. What is the minimum magnitude of completeness in the catalog? Will the missing events affect your results (such as the exponent in TPL)?

2. Line 336, you mentioned about space and time, but you only showed results for different temporal blocks. May you need to give some results using different spatial blocks (divide the study region in Fig.3a into several sub-regions)

3. How would you results benefit earthquake forecasting?

You may need to reference the following two papers, which are important pioneer work about scaling law for earthquakes.

1. Unified Scaling Law for Earthquakes, P H Y S I C A L R E V I E W L E T T E R S, 2002;

2. Unified scaling law for earthquakes, PNAS, 2002

---

## Author Response (AR1)

Ms. No.: nhess-2018-315 Title: A Taylor's power law in the Wenchuan earthquake sequence with fluctuation scaling

General: This manuscript applied Taylor's power law (TPL) into the released energy of earthquakes in the Wenchuan aftershocks. It confirms the existence of TPL in earthquake sequence, i.e. the variance is shown as a power law function of the mean. TPL holds for different time spans, although the intercept values of the linear regressions increase with the increase of time spans. I think this is a good paper that is well written and scientifically interesting. I applaud the general approach and central research question of the paper. I did not find any major problem. Here I listed my suggestions which may help in improving the manuscript.

1. I suggested incorporating section '2 Taylor's power law' into 'Introduction', and adding a new section 'Data source and processing method' (or similar subhead) which includes section '1 Wenchuan earthquake sequence' and part of section '3 Data processing method and results'. Finally I would expect four sections in the main text: 'Introduction', 'Data source and processing method', 'Results', and 'Discussion and conclusions'.

It is done as the reviewer has suggested but with a little difference. Changed Sections are in red in the revised version of the paper.

2. I think lines 181-182 can be deleted as they are duplicated with lines 178-179. It is done.

3. It is not clear for me why the authors placed Figures 5 and 6 into Discussion but not Results. Would it be better to move them into Results?

We have put Figures 5 and Figure 6 and corresponding words into Section results. We also add some words in red into the revised version of the paper.

4. It is interesting that the authors showed the positive relationship between the intercept and the interval (Figure 6). Inspired by them, I drew the relationship between the slope and the interval (see attached figure, panel a). Furthermore, the interval shows strong negative relationship with the slope (panel b). This is interesting and I have no idea to explain them. I understand this may be out of the scope of the current manuscript, so the authors do not need to discuss them here. This may be helpful in their future work.

We thank the reviewer for the interesting work and figures! Yes, it is important to do further investigation on this topic in the near future.

**Anonymous Referee #2**

**A Taylor's power law in the Wenchuan earthquake sequence with fluctuation scaling by Peijian Shi et al.**

The authors explore Taylor's power law (TPL) of the released energy of Wenchuan earthquakes sequence on varying timescales. In my opinion the paper is interesting and well written. Here there are a couple of major comments:

- The authors show that the exponent of the TPL is a value independent of timescale equal or almost equal to 2. Why does it happen? The authors mention some general explanations, some of them related

to Ecology, but none of them is focused on the released energy of earthquakes along time. In other works, my question is, for what physical reason would the exponent of the TPL be equal to 2?

Considering the previous investigations, we think the physical reason of the slope b=2 in TPL during our work is that, after the main shock, there is a stable spatial-temporal dependent energy release caused by regional stress adjustment and redistribution during the fault revolution after the main shock. The related words are also added to the revised paper in red.

- Note that using the logarithm to model the intercept as a function of the temporal block size A, the overall formula of the TPL remains,

 $V = kA\beta\mu b$

This is an interesting expression that would deserve more attention in the paper. In particular, the exponent of A (it is around 0.2) is an interesting parameter that should be highlighted and discussed in the paper.

We have added something respectively to Section results and discussion in red.

**Minor comments**

- Lines 20-22: "On the other hand...approximately definite and deterministic". This paragraph is not clear, what does it mean?

- I realize that the authors are using along the paper logarithms with base 10, it should be mentioned. Moreover, notation "lg" for logarithms is not standard and it would be better changed to "log" along the paper.

We have changed "lg" into " $\log_{10}$ " in revised paper.

- Usually, when the mean is denoted with the Greek letter  $\mu$  the variance is denoted as  $\sigma 2$ . We have changed "V" into " $\sigma^2$ " in revised paper.

- Lines 296-298: "The variations of the estimated exponent b... are shown in Figure 6". Something is wrong here because Figure 6 shows the variation of the estimated intercept.

Yes and the reviewer is right. We have changed the sentence "The variations of the estimated exponent b of the Wenchuan sequence... are shown in Figure 6" into "The variations of the estimated intercept of the Wenchuan sequence... are shown in Figure 6".

**Anonymous Referee #3**

Received and published: 14 February 2019

Earthquake forecasting is very important but challenging. Due to the inherent randomness and complexity of rupture process, the forecast can only be made in probabilistic manners rather than in the form of deterministic predictions. Therefore, statistical methods play an important role in earthquake forecasting and hazard assessment.

Taylor's power law (TPL) has been widely testified across space and time in biomedical sciences, botany, ecology and many other fields. The current manuscript shows its application in the study of Wenchuan earthquake sequence. The results suggest that the mean–variance relationship of the energy release from the earthquakes could be predicted and the exponent of TPL are approximately 2.

The manuscript is well organized. The discussion is thorough and the methods are solid. I recommend that the manuscript could be considered for publication after replying the following questions.

1. What is the minimum magnitude of completeness in the catalog? Will the missing events affect your results (such as the exponent in TPL)?

The catalog of the Wenchuan earthquake sequence includes all events with the magnitude M > 0, which has been labeled in red in the paper. The missing events can lead to the exponent in TPL increase, for example, the estimated exponent  $b \sim 2.1-2.2$  when the magnitude of the events used is with M > 1.0. We also added some words to the revised paper in red.

2. Line 336, you mentioned about space and time, but you only showed results for different temporal blocks. May you need to give some results using different spatial blocks (divide the study region in Fig.3a into several sub-regions)

In fact, the aftershock area is changing as the time span changes during our calculations because different events occurred in different locations. As for dividing the total aftershock area into several sub-regions, it will be our consideration in near future because we now find not enough evidence on how to divide this region. This problem is also mentioned in Section discussion and conclusions of the paper in red.

3. How would you results benefit earthquake forecasting?

You may need to reference the following two papers, which are important pioneer work about scaling law for earthquakes.

1. Unified Scaling Law for Earthquakes, PHYSICALREVIEWLETTERS, 2002;

2. Unified scaling law for earthquakes, PNAS, 2002

We thank the reviewer for the interesting problem and valuable references!

Earthquake forecasting is a complex problem and it depends on many factors. We do not explicitly discuss this problem in the paper although our results show some law of energy release during the Wenchuan earthquake sequence. But we tend to think that most of statistical results may finally attribute to physical mechanism of earthquakes, for example, nonzero driving force in the crust of the Earth as mentioned in references listed above or regional stress adjustment and redistribution we have mentioned in the paper. Of course, we also cite some results attained by Bak et al. (2002) and Christensen et al. (2002) and add the references into the reference list of revised paper.

---

## Author Response (AR2)

**Editor Decision: Publish subject to minor revisions (review by editor)** (10 Apr 2019) by Oded Katz

Comments to the Author:

Dear Mei Li,

The two reviews given to your revised version of the manuscript indicated that the manuscript is ready for publication.

However, I will ask you, before accepting the manuscript, to separate the Discussion and the Conclusions parts. I.e. Conclusions in a separate paragraph from line 378 and modification as needed to make sure that both paragraphs are complete.

We thank the editor and now Discussion and Conclusions are two separated sections in the revision of the paper.